# Silver Nanoparticles: Bactericidal and Mechanistic Approach against Drug Resistant Pathogens

**DOI:** 10.3390/microorganisms11020369

**Published:** 2023-02-01

**Authors:** Pragati Rajendra More, Santosh Pandit, Anna De Filippis, Gianluigi Franci, Ivan Mijakovic, Massimiliano Galdiero

**Affiliations:** 1Department of Experimental Medicine, Section of Microbiology and Clinical Microbiology, University of Campania “L. Vanvitelli”, Via De Crecchio, 7, 80138 Naples, Italy; 2Systems and Synthetic Biology Division, Department of Biology and Biological Engineering, Chalmers University of Technology, 41296 Gothenburg, Sweden; 3Department of Medicine, Surgery and Dentistry, Scuola Medica Salernitana, University of Salerno, 84081 Baronissi, Italy; 4Novo Nordisk Foundation Center for Bio Sustainability, Technical University of Denmark, 2800 Kongens Lyngby, Denmark

**Keywords:** multidrug-resistant bacteria, silver nanoparticles, biofilm forming organism

## Abstract

This review highlights the different modes of synthesizing silver nanoparticles (AgNPs) from their elemental state to particle format and their mechanism of action against multidrug-resistant and biofilm-forming bacterial pathogens. Various studies have demonstrated that the AgNPs cause oxidative stress, protein dysfunction, membrane disruption, and DNA damage in bacteria, ultimately leading to bacterial death. AgNPs have also been found to alter the adhesion of bacterial cells to prevent biofilm formation. The benefits of using AgNPs in medicine are, to some extent, counter-weighted by their toxic effect on humans and the environment. In this review, we have compiled recent studies demonstrating the antibacterial activity of AgNPs, and we are discussing the known mechanisms of action of AgNPs against bacterial pathogens. Ongoing clinical trials involving AgNPs are briefly presented. A particular focus is placed on the mechanism of interaction of AgNPs with bacterial biofilms, which are a significant pathogenicity determinant. A brief overview of the use of AgNPs in other medical applications (e.g., diagnostics, promotion of wound healing) and the non-medical sectors is presented. Finally, current drawbacks and limitations of AgNPs use in medicine are discussed, and perspectives for the improved future use of functionalized AgNPs in medical applications are presented.

## 1. Introduction

Nanomedicine is a field of science that deals with the manipulation and fabrication of nanoparticles [1]. These particles are found in the range of 1–100 nm, along with the unique biomedical properties that differentiate them from the bulk elemental form [2,3]. The small size of NPs gives them a larger surface-to-volume ratio, making them more beneficial in biochemical and catalytic activity than other particles with the same composition [1,4]. They have been successfully employed in different areas such as drug delivery, biomedical sciences, chemical industries, wastewater treatments, waste disposal treatments, gene delivery, optics, mechanics, etc. [5]. Due to their potent antimicrobial and anti-inflammatory effect, they have become more popular and attracted significant attention. To inhibit bacterial infections, AgNPs have been used in various physical, chemical, biological, and pharmaceutical fields; for instance, cream, moisturizers, face mask or ointments to avoid opportunistic infections [6]. Although AgNPs are used in various sectors, the mechanism of particle formation still needs to be fully uncovered.

Traditionally, AgNPs are synthesized by physical and chemical methods to produce NPs with colloidal dispersion, stable size, and shape in water or other organic solvents [7,8]. Nevertheless, the biological synthesis of nanoparticles is widely explored due to toxic substances, laser ablation, hydrothermal synthesis, pyrolysis, and inter-gas condensation [9,10]. The biological method provides high yield, stability, and, most important, less toxicity, making them superior to use as compared to NPs synthesized by physical and chemical processes. The extraordinary features of biosynthesized silver NPs as antimicrobial, antifungal, antiviral, and anticancer agents attract the attention of researchers.

Nanosilver used in a real-life application in the case of wound-dressing material nanocrystals is presently enforced to inhibit bacterial infections in open wounds [11]. They are also applied on the surface of the catheter to avoid biofilm formation and to keep the surface free from contaminants. Based on silver ion substances, the bactericidal effect of silver-holding materials has been frequently discovered. They target cytoplasmic membranes and nuclei acids; they suppress the respiratory chain enzymes and alter the membrane permeability [12]. Various studies support the finding of nanosilver contact with the bacterial membrane by providing electron microscopy analysis [13]. Xu et al. studied the number of silver-hold bactericidal NPs via dark-field microscopy. However, the general commercial use of NPs creates a tone of nano waste that harms humans and the environment [14,15]. This study broadly explains the mechanism behind NP synthesis and their bactericidal mechanism against different bacteria and biofilm-forming organisms. Finally, we highlight their real-life application with a few disadvantages and alternative solutions to those issues.

## 2. Silver Nanoparticles and Antibacterial Activity

Antibiotics are the primary source of therapy against all bacterial infections. However, multidrug-resistant organisms are becoming a global threat. Infections caused by MDR pathogens are difficult to treat with the currently available antibiotics. Developing new antibiotics is time-consuming and will require tremendous economic investment. Therefore, nanotechnology and silver nanoparticles are considered alternative sources to overcome the threat of antimicrobial resistance [16].

Top-down and bottom-up methods are primarily utilized to synthesize NPs [17]. In top-down approaches, it involves breaking bulk material into nanosized or nanomaterials, along with significant structures and their original characteristics, without changing the atomic or sub-atomic level [18]. The bottoms-up method involves the chemical or physical forces to build the nanoscale structures by assembling or self-assembling processes [18]. However, the bottom-up approach is further classified into physical, chemical, and biological subgroups. Figure 1 explains in detail the diagrammatic representation of the synthesis process.

The physical approach of NPs synthesis is generally a robust method of evaporation and rapid condensation, performed by a tube heater and climatic load [19]. However, the physical approach presents some limitations, including substantial space for tube heater and incredible utilization vitality for enhancing the ecological temperature across the base material. Therefore, to overcome this, different strategies for the amalgamation of AgNPs because of the physical approaches have been produced [20,21,22,23]. As an alternative to this method, researchers have used an economical way to prepare nano-silver colloidal dispersion in water or organic solvent by the so-called wet chemical route. This approach includes chemical reduction, microwave-assisted synthesis, microemulsion, photoreduction, electrochemical, etc. [5,24,25,26]. The chemical method of synthesis can be classified into chemical reduction, electrochemical, irradiation-assisted chemical, and pyrolysis methods [26,27]. The synthesis of AgNPs requires metal precursors, reducing agents, and stabilizing or capping agents. Ascorbic acid, alcohol, borohydride, sodium citrate, and hydrazine compounds are widely used as efficient, reducing agents [28,29].

The shortcomings associated with the physical and chemical synthesis strategies for developing AgNPs include expensive techniques and harmful, perilous chemicals, which might cause impending ecological and biological hazards. Therefore, there was a need to establish methods to overcome these limitations and produce stable and therapeutically active AgNPs. This is how the biological method of processes of NPs has emerged. NPs are synthesized with different biological entities: plant extract (root, leaves, stem, fruits) and microorganisms (yeast, bacteria, fungi). In the biological extract, amino acids, proteins, vitamins, polysaccharides, enzymes, and phytoconstituents act as reducing agents [30].

The biological synthesis of NPs includes two types of synthesis mechanisms: one is extracellular, and the other is intracellular. The process of extracellular synthesis involves microbial enzymes and protein, bacterial or fungal cell wall components, or any other organic content in media. At the same time, the intracellular process consists of the interaction of different carboxyl and amine groups with metal ions. Irrespective of the synthesis of NPs by either intra- or extracellular process, the critical factor in the synthesis process is enzymes. NADH-dependent nitrate reductase has a lead role in AgNP synthesis. It acts as an electron shuttle, taking electrons from nitrate molecules and transforming them into metal ions leading to NPs formation. Researchers also believe that the carboxylic group in bacterial cell walls carries a negative charge that may provide an electrostatic interaction between these groups. In Ag ions or nanocrystal reduction, a few amino acids are involved: arginine, aspartic acid, cystine, glutamic acid, lysine, and methionine. They act as catalysts and produce hydroxyl ions that react with the reducing agent’s aldehyde. [31]

The primary mechanism suggested in plant-based NPs synthesis are:Studies support the finding that the bioreduction of metal ions occurs due to the presence of protein, which traps the metal ions, and reduction occurs. This leads to change in the secondary structure of protein and formation of metal ions seed/nuclei and helps in the construction of NPs.The most accepted approach is a plant extract containing various phytochemicals. Based on the literature data available, they support that not one particular active ingredient or phytochemical is responsible for reducing NPs. Nevertheless, the various other plant components and secondary metabolites also play an essential role. Some active compounds included multiple proteins, enzymes, amino acids, vitamins, polysaccharides, polyphenols, alkaloids, flavonoids, and organic acids [32].

During the process of NPs formation, the metal ions form a complex with the phenolic hydroxyl compounds of biomolecules. Oxidation of biomolecules, metal ions reduce to a zero-oxidation state; further, it helps stabilize particles by forming capping on the surface. Hence, it is not required to put any other reducing or capping agent during the synthesis of NPs. In most studies, it is observed that the biobased reduction of AgNPs reduces toxicity by forming the biological corona on the NP’s surface. However, in some cases, it is also observed that the biological extract exhibits some toxic effects. For this reason, researchers advise checking the toxicity effect of biological extract alone and after synthesizing the AgNPs. [33,34]. Their potential antimicrobial activity not only makes them a potent candidate to be used as an alternative to available drugs against MDR pathogens, but they also reduce the unwanted toxicity to mammalian cells. Table 1 collectively represents the green synthesized AgNPs and their antimicrobial activity. The past few years’ integrations of green science systems and philosophies into nanotechnology are awesomely intriguing.

### 2.1. AgNPs and Their Mechanism of Action against Bacteria

Silver is positively charged and thus tends to react with negatively charged biomolecules such as phosphorous and sulfur, which are the main components of the cell membrane, proteins, and DNA bases. Silver nanoparticles damage the cell wall and membrane of bacterial cells, causing various morphological changes [60]. Several studies have shown the effective use of silver nanoparticles against Gram-positive and Gram-negative pathogens [61]. The critical characteristics of nanoparticles are a size that must be in the field of 1–100 nm, and they also possess an excellent surface volume ratio and shape of the nanoparticles. All of these factors play a vital role [62]. The size of the nanoparticles plays an important role in antibacterial activity. The various studies demonstrates that the smaller the dimension of the nanoparticle, the greater is the ability to penetrate the bacteria [63,64,65,66].

The exact reason for the antimicrobial effect of AgNPs on bacteria has yet to be cleared. In Figure 2, we explain a possible mechanism of action of AgNP by which it paves antimicrobial activity. Silver ions are continuously released from silver nanoparticles, which may consider a mechanism for killing microbes. Silver ions can easily adhere to the cell wall and cytoplasmic membrane as they are more closely related to sulfur proteins and also due to electrostatic attraction [54,67,68,69,70,71,72]. At the same time, the bacterial envelope is disrupted because when silver ions attach to the cell wall or cytoplasmic membrane, it enhances the permeability of the cell and ultimately leads to cell disruption. When free silver ions are uptake by cells, they deactivate respiratory enzymes, generating reactive oxygen species interrupting adenosine triphosphate production. ROS is the principal species that provokes the activity of DNA modification and cell membrane disruption. In DNA, sulfur and phosphate are essential components. Still, the interaction of AgNPs with sulfur and phosphorus in DNA can cause difficulties in DNA replication and cell reproduction or even result in the termination of bacteria. Sometimes, the ribosome’s denaturation in the cytoplasm occurs as the silver ions can inhibit protein synthesis [73,74].

AgNPs garner in the pit once they are attached to the cell surface [75], and a cell membrane’s denaturation occurs due to the accumulation of AgNPs. They also can enter the cell wall and alter the structure of the cell membrane due to their nanoscale size [75]. Denaturation of the cytoplasmic membrane also causes cell rupture leading to cell lysis. AgNPs are also involved in bacterial signal transduction. Phosphorylation of protein substrate and nanoparticles can dephosphorylate tyrosine residues on peptide substrates, ultimately affecting the signal transduction in bacteria. Disruption in signal transduction can escort cell apoptosis and termination of cell multiplication [54,76]. The antibacterial activity and mechanism of AgNPs can be affected due to dissolution status in exposure media. Intrinsic AgNP characteristics and surrounding media are some synthetic and processing factors on which dissolution efficiency is dependent [77]. The presence of organic and inorganic components in media affects the dissolution properties of AgNPs by causing aggregation of particles or complexing with silver ions. The Ostwald–Freundlich equation described in theoretical terms that the particle size and shape can influence the release of silver ions. Due to the large surface area, smaller AgNPs with the spherical or quasi-spherical format are more prone to silver release [77]. Such dissolution behavior can be avoided with the help of capping agents by modifying the surfaces of AgNPs [78]. A recent study proves that AgNPs release silver ions faster in an acidic solution than in a neutral solution [79].

The antimicrobial efficiency of AgNPs is more against Gram-negative than Gram-positive bacteria [80]. Gram-negative bacteria have a thick LPS layer in their cell wall with a thin peptidoglycan layer. In contrast, Gram-positive bacterial cell walls contain thin LPS and thick peptidoglycan layers. The cellular wall made of a thick peptidoglycan layer reduces the chances of AgNP penetration into cells [80]. The uptake of AgNPs is significant for the antibacterial effect, as it has been demonstrated on various Gram-positive and Gram-negative bacteria [78]. In Figure 3 and Table 2, we explain the comparative mode of action of AgNPs on both Gram-positive and Gram-negative organisms. It is acknowledged that AgNPs smaller than 10 nm can alter cell permeability by entering into bacterial cells and causing cell damage.

The physical properties such as shape, size, surface charge, dosing, and diffusion state can influence the AgNPs’ antimicrobial effects [81,82,83,84,85]. According to different reports, it is suggested that the relatively small NPs smaller than 10 nm can penetrate easily inside the bacterial cell and ultimately cause cell lysis. The biofilm formation and other in vivo environments hamper the transport of Ag+ ions or AgNPs. Bacteria in biofilm get protected from the effect of Ag+ ions and AgNPs [59]. The interaction of Ag+ ions with biological macromolecules such as enzymes and DNA based on electron discharge or free radical generation are some other mechanisms of action of Ag+ ions or AgNPs for the deactivated [86]. Alternation in protein synthesis and cell wall synthesis, evidenced by amassing of enveloped protein precursor or disruption of the outer cellular membrane, leading to ATP leakage, are also demonstrated to pave a significant role in the antimicrobial activity of AgNPs [87].

In addition, the role of nanoparticle shape concerning their antimicrobial potential has also been demonstrated. Ali Bakhtiari-standard et al. synthesized AgNPs in spherical shape shows excellent antimicrobial activity. They tested NPs against *E.coli* and *S. aureus*. *E.coli,* and they observed that *E.coli* is more susceptible than *S. aureus*. [88]

Numerous studies have explained the importance of the shape of NPs. As the shape of NPs changes, the potential antimicrobial effect also changes [73,89]. The interaction of AgNPs on bacteria, viruses, and fungi is shape-dependent [84,90,91,92,93]. The results obtained from the electron microscope revealed shape-dependent cell membrane disruption of Gram-negative *E. coli* upon treatment of differently shaped NPs [15] [94]. It is also reported that the AgNPs with rod and spherical shapes possess sound antimicrobial effects [95], while Ja Young Cheron et al. proved the importance of the shape of NPs. They synthesized NPs in three different shapes spherical, disc, and triangular. The highest bactericidal effect was observed against spherical > disc > triangular NPs. It might be due to the indifference to the release of silver ions because of differences in the surface area of AgNPs. Therefore, they concluded that morphological changes could control the AgNPs’ activity. Another study conducted by A. Shareef synthesized spherical and octahedral-shaped NPs. They found octahedral NPs have a bactericidal effect, while spherical shape NPs show a bacteriostatic effect. This presumes that the indifference effect in antimicrobial effect is due to the shape. As the shape of NPs changes, the efficiency of NPs changes, which might be due to the higher surface area of octahedral AgNPs, compared to spherical AgNPs [65]. Wen-RuLi et al. examined the activity of AgNPs against *P. aeruginosa*, *S. epidermidis*, and *E. coli* and demonstrated that AgNPs are more active against *E.coli* than *P.aeruginosa* and *S. epidermidis* [96]. From the various studies on MDR bacteria, AgNPs are effective against those pathogenic bacteria such as *E. coli*, *S. Typhi*, *S.epidermidis* and, *S. aureus*, *P. aeruginosa* [97].

Researchers have investigated the combination and individual antibacterial activities of five conventional antibiotics with NPs, where the activity of the nanoparticles was checked against eight different MDR bacterial species by the disc diffusion method [98]. The antimicrobial activity was demonstrated by the zone of inhibition around the disc, loaded with antibiotics alone, AgNPs, and their conjugates. The effect of various combinations such as ciprofloxacin + AgNPs [99], Imipenem + AgNPs [100], gentamycin + AgNPs [101], vancomycin + AgNPs [102], AgNPs+ Imipenem [103], trimethoprim + AgNPs [103], ciprofloxacin + AgNPs [104], gentamycin + AgNPs [105], vancomycin AgNPs [106] and trimethoprim + AgNPs were examined against various pathogens. The synergistic effect of antibiotics and nanoparticles resulted in zone inhibition around the disc, while the zone size was around 0.2–2.8 cm (average, 2.8 cm). These antibiotics were used in combination with nanoparticles to improve the potency of a drug. The synergistic action of AgNPs and antibiotics demonstrated the enhanced antibacterial effects; therefore, the simultaneous action of antibiotics and AgNPs can hamper the resistance development by pathogenic bacteria because of the reduced amount of antibiotics administered. There are various studies currently under clinical trial despite all the plus and minus effects of AgNPs. Table 3 collectively gives an idea regarding the antimicrobial effect of AgNPs against bacteria. While Table 4 indicates the activity of AgNPs against biofilm forming microorganisms.

**Table 2 microorganisms-11-00369-t002:** Selected studies of NP’s mode of action against Gram-positive and Gram-negative bacteria.

Bacterial Strain	Nanoparticles	Mode of Action	References
*Staphylococcus aureus*	AgNPs	Inhibits the respiratory chain dehydrogenase Interfere with the bacterial growth and metabolism of the cell	[107]
*Escherichia coli* *Salmonella typhimurium*	AgNPs	Disrupts the integrity of Gram-negative bacteria by Depolarization and destabilization of membrane	[107]
*Pseudomonas aeruginosa*	AgNPs	AgNPs generate free radicals that damage the cell membrane	[108]
		ROS interacts will the cell wall and cell membrane	
*Pseudomonas aeruginosa* PAO1	AgNPs	Attach to the cell membrane surface and disrupt its permeability	[109]
*Serratia proteamaculans* 94	AgNPs	By modifying the cell potential and inhibiting cell respiration	
*Escherichia coli* ATCC25922 *Staphylococcus aureus* ATCC25923	AgNPs	Destruction of cell membrane and rise of ROS Concentration	[110]
*Proteus* spp.	AgNPs	AgNPs forms pits in the cell wall of bacteria, enter the periplasm And destroy the cell membrane. Degradation and loss of DNA	[111]
*Klebsiella*	AgNPs	Replication which inhibits bacterial growth	
Multidrug resistant *P. aeruginosa* Ampicillin resistant *E. coli* 0157:H7 Erythromycin resistant *Streptococcus pyogenes*	AgNPs	Inhibits cell wall synthesis, nucleic and synthesis protein Synthesis mediated by 30S ribosomal subunit	[112]
*Vibrio cholera*	AgNPs	Penetrating in the bacterium disrupts its functions and releases silver ions that affect the antibacterial activity.	[113]
*Escherichia coli*	AgNPs	AgNPs anchor and penetrate to bacterial cell wall	[113]
*Salmonella typhi* (multidrug resistant)	AgNPs	Modulate cellular signaling by putative dephosphorylating key	
*Staphylococcus aureus*	AgNPs	Peptide substrates on tyrosine residues	
*Pseudomonas* *aeruginosa* Gram-negative	AgNPs	Interaction with ROS and attachment of AgNPs at the microbial cell wall	[114]
*Escherichia coli* AB1157 Gram-negative	AgNPs	Damage the cellular DNA by influencing the base excision repair system	[115]
*Staphylococcus aureus* ATCC25923 Gram-positive	AgNPs	Destruction of microbial cell membrane and rise of ROS concentration	[109]
*Escherichia coli* *ATCC25922* Gram-negative *Escherichia coli DH5_*Gram-negative	AgNPs	Accumulation of AgNPs in the cell wall and cell membrane of bacterial cell	[116]
*Bacillus* Calmette-Guérin Acid-fast Gram-positive Multidrug resistant *Escherichia coli (MC-2)* Gram-negative	AgNPs	Disruption of the cell membrane through Multidrug resistant formation of ROS	[117]
*Staphylococcus aureus* *(MMC-20)* Gram-positive *Proteus* sp. Gram-negative	AgNPs	The cell wall ruptured and inhibited DNA replication, thus inhibiting bacterial growth.	[118]
*Klebsiella* sp. Gram-negative *Staphylococcus aureus* Gram-positive	AgNPs	Oxidative stress causes alteration in kynurenine protein. Activation of kynurenine pathways thus inhibits bacterial growth.	[32]
Gram-negative bacteria	AgNPs	Binding to the cell wall and penetrating it; modulation of cellular signaling	[15]
*Escherichia coli*	AgNPs	Damage of bacterial cell membrane in multiple locations, formation of irregular pits	[119]
*Escherichia coli*	Nano ag	Changes in expression of genes encoding envelope proteins (accumulation of envelope protein precursors), destabilization of the outer membrane, disturbance of proton motive force	[13]
*Escherichia coli*	AgNPs	Damage of membranes, incorporation of silver nanoparticles into membranes, forming pits, disturbances in permeability	[120]
Gram-positive and Gram-negative bacteria	AgNPs	Binding to the cell membrane, permeability changes, disturbances in the respiration process, penetration of the bacterial membranes, interaction with DNA, releasing Silver ions	[121]
*Escherichia coli*, *Klebsiella pneumonia*, *Bacillus pumilus* and *Staphylococcus aureus*	Chitosan-AgNP	Not specified	[122]
*Acinetobacter* *baumannii*, *Escherichia coli*, *Pseudomonas aeruginosa* and *Salmonella* *enteric*	AgNPs	Not specified	[123]
*Escherichia coli*	GO-L-cys-AgNPs	Damages to the cell membrane	[122]
Gram-positive and Gram-negative bacteria	AgNPs	Not specified	
*Escherichia coli*, *S. typhus*	AgNPs	Anchor to the cell membrane, perforation formation in the membrane results in cell lysis	[124]
*S.epidermidis*, *Staphylococcus aureus*, *Enterococcus faecalis*	Ag colloid-NPs (various saccharides as reducing agent)	Proposed mechanism: attach to the cell membrane, disturb its permeability and respiration, penetrate the bacteria, Ag colloid-NPs, and its releasing silver ions react with bacterial DNA.	[91]

**Table 3 microorganisms-11-00369-t003:** Different studies under the clinical trial phase (Clinical trials.gov).

Sr. No	Study Name	Phase of Study	Identifier Number
1.	Topical Application of Silver Nanoparticles and Oral Pathogens in Ill Patients	Completed	NCT02761525
2.	Efficacy of Silver Nanoparticle Gel Versus a Common Antibacterial Hand Gel	Recruiting	NCT00659204
3.	The Antibacterial Effect of Nanosilver Fluoride on Primary Teeth	Completed	NCT05221749
4.	Radiographic Assessment of Glass Ionomer Restorations with and Without Prior Application of Nano Silver Fluoride in Occlusal Carious Molars Treated with Partial Caries Removal Technique	Completed	NCT03193606
5.	Antibacterial Effect of Nano Silver Fluoride vs. Chlorhexidine on Occlusal Carious Molars Treated with Partial Caries Removal Technique	Completed	NCT03186261
6.	Silver Nanoparticles in Multidrug-Resistant Bacteria	Completed	NCT04431440

**Table 4 microorganisms-11-00369-t004:** Selected studies of NP’s mode of action against Gram-positive and Gram-negative bacteria.

Sr. No	Study Name	Phase of Study	Identifier Number
1.	Nanosilver Fluoride to Prevent Dental Biofilms Growth (NSFCT)	Completed	NCT01950546
2.	Effect of Metallic Nanoparticles on Nosocomial Bacteria	Recruiting	NCT04775238
3.	Silver Nanoparticle Investigation for Treating Chronic Sinusitis (SNITCH)	Withdrawn (IND not approved)	NCT03243201
4.	The Effectiveness of Topical Silver Colloid in Treating Patients with Recalcitrant Chronic Rhinosinusitis (CRS)	Completed	NCT02403479

### 2.2. Biofilm

Biofilm formation is of great concern as the community of the cells in consortia is widely known to show more antibiotic resistance than a free-living multicellular organism [125]. Bacteria possess high antimicrobial resistance once they form biofilms on the biotic and abiotic surfaces. It has been reported that bacterial cells in biofilms are one thousand times more resistant to antibiotics and disinfecting agents than planktonic cells [126,127]. Biofilms are the primary cause of nosocomial infections and chronic illnesses, which are difficult to resolve with antibiotics and reoccur persistently. One of the significant examples of biofilm-driven problems is chronic respiratory illnesses associated with cystic fibrosis [126]. *Pseudomonas aeruginosa* is the most commonly studied opportunistic biofilm-forming pathogen, which leads to lung damage in immunocompromised patients [128]. Endocarditis, chronic bacterial prostatitis, and otitis media are caused by bacterial biofilm [129,130,131]. Bacterial biofilm also causes oral diseases such as dental periodontitis and caries [129,132].

Biofilm forms a colonial layer on medical devices such as urinary catheters, prosthetic heart valves, surgical sutures, contact lenses, and dental sutures, which cause recurring health problems in individuals [127,129,132]. Furthermore, biofilm has also been seen to be causing foodborne illnesses [133,134]. Biofilm forms a layer on the surfaces of poultry and meat products [133,135,136,137]. *Escherichia coli*, *Listeria monocytes,* and Salmonella spp. have the more remarkable ability to form a biofilm, which causes foodborne illnesses and is still a significant challenge within the food industry [138,139,140,141].

Despite the environment, all biofilms share a common characteristic: the synthesis of an extracellular polymer matrix. This extracellular polymer matrix material comprises polysaccharides, proteins, and nucleic acids produced by biofilm-producing microorganisms [142]. The EPS attaches the biofilm cells firmly to the surfaces and protects them from harsh conditions. The protective nature of EPS is a substantial factor that mediates the attachment of bacteria to their substrate, which in turn causes the ineffectiveness of antibiotics and the eradication of biofilm-related diseases [142]. Thus, biofilm plays an essential role in the pathogenesis and the development of severe clinical implications [142,143]. Biofilm is a defense mechanism of bacteria that protects them from extreme environmental conditions, toxicity to heavy metals, high temperature and pressure, ultraviolet radiation (UV), and from penetration of antibiotics into the cell surface [143,144].

The first stage of biofilm occurs when a specific environmental signal induces a genetic program in free-living planktonic cells. In response to this signal, the planktonic cells attach to nearby surfaces utilizing flagella, pili or lipopolysaccharides. The attached cells begin to coat the organic monolayer of polysaccharides or glycoproteins to which more planktonic cells can link. When the cell enters the stage of biofilm formation, it no longer maintains its flagella and moves along the surfaces using twitching motility. Ultimately, it stops moving and binds firmly to the surface. As the cells attach to the surface and divide, they form microcolonies and communicate with each other by sending and receiving chemical signals called Quorum sensing [145,146]. When the chemical signals reach a specific concentration that the cells can sense, this concentration triggers genetically regulated changes that cause the cells to bind to each other tenaciously [147,148].

#### 2.2.1. Mechanism of Action of AgNPs against Biofilm

The biofilm formation is initiated by adherence to cell surfaces, but when the cells are treated with AgNPs, they fail to adhere and build such a community on the characters. It has great importance, especially during the fight against biofilm formation pathogenic organisms [149,150]. The previous report showed that AgNPs contribute to neutralizing adhesive substances involved in biofilm formation [151]. In addition, Jena et al. suggested that by disrupting the bacterial actin cytoskeleton network, AgNPs can mediate apoptosis of the bacterial cell [152]. NP affects the actin cytoskeleton MreB, causing morphological changes in the shape of bacteria, which increase the fluidity in the membrane and cause the rupture of the cells. At the same time, the mass of AgNPs on the cell membrane affects bilayer integrity and appears broken.

In contrast, entering AgNPs directly into the cell and interacting with vital organelles ultimately leads to cell death. Silver ions interact with the disulfide bond of enzymes responsible for cellular metabolism and thiol group, indeed respiratory enzymes, and deactivate them by generating reactive oxygen species (ROS). A double bond of fatty acid can be oxidized due to ROS in the membrane, which concedes for the generation of other free radicals and damages the cell membrane [153,154]. In Figure 4 we have explained the detailed mechanism of action of AgNPs in diagrammatic format.

The shape and structure of cellular enzymes change upon exposure to AgNPs because the silver catalyzes the formation of disulfide bonds responsible for these cell membrane changes. AgNPs’ treatment was found to affect the expression of vital proteins and enzymes such as 30S ribosomal subunit, succinyl coenzyme A synthetase, maltose transporter (MalK), and fructose bisphosphate aldolase. Due to the binding of silver ions to the 30S ribosome subunit, protein synthesis can be altered by deactivating the ribosome complex. The effect of AgNPs on transcription and translation ultimately leads to cell death. The AgNPs are also demonstrated to influence the essential enzymes such as succinyl coenzymes-a-synthetase involved in the tricarboxylic acid cycle, creating disturbances in cellular metabolism [155]. It is also reported that the bactericidal properties of AgNPs were associated with disrupting bacteria RNA transcription, purines, pyrimidines, and fatty acids [75]. The interaction of AgNPs with cells can affect DNA replication [77,110,156]. AgNPs broke the H bond between anti-parallel base pairs as it formed a complex with the nucleic acid. AgNPs are also involved in changing the molecular state of DNA from relaxed to condensed form. As a result, replication ability is decreased [157,158,159].

The phosphorylation cycle and dephosphorylation cascade in the microbial cells represent necessary signals for microbial growth and cellular activity. Due to AgNPs’ treatment, microbial growth is inhibited by dephosphorylating tyrosine residues on crucial bacterial peptide substrates [113]. AgNPs produced by a green method were found to trigger the activation of the p53 protein, which also acts as a suppressor of malignant tumor formation in the mammalian cell, as well as caspase3, which plays a significant role in cellular apoptosis [151]. In this regard, the relationship between the antimicrobial effect and the size of nanoparticles is well established. As the size of nanoparticles is tiny, they will increase the interaction between the cell surface, and increase the penetration inside the bacteria [160].

#### 2.2.2. Can Nanoparticles Affect Biofilm Formation?

It is reported that AgNPs interact with the bacterial cell surface and disrupt the cell membrane’s permeability, inhibiting the respiratory enzymes and producing reactive oxygen species (ROS). In a study conducted by Soo-Hwan K. et al., the antibacterial activity of silver nanoparticles was studied against Staphylococcus aureus and Escherichia coli. The MIC of silver nanoparticles against *S. aureus* and *E. coli* was 100 µg/mL. The growth curves of bacterial cells with different concentrations of AgNPs were compared, where the bacterial cells were treated with 50 µg/mL, 100, and 150 µg/mL. Inhibition was seen in the bacterial cells treated with 100 and 150 µg/mL within 4 h. In addition to the bacteriostatic effect, changes in the morphology of bacterial cells were also observed. The bacterial cells treated with AgNPs show damage in the cell membrane, and fragments were formed due to damage to the cell membrane [161]. Similarly, in another study, AgNPs particles of 8.3 nm in diameter affected the growth of Pseudomonas aeruginosa PAO1 biofilms at various concentrations of 4–5 µg/mL, ten µg/mL, and 20 µg/mL [19]. The antibiofilm activity of AgNPs has been demonstrated in several studies. The following research shows antibiofilm activity of β1-3 glucan-binding protein-based silver nanoparticles was assessed [17].

Navindra et al. demonstrated the effect of AgNPs against the biofilm of *Pseudomonas aeruginosa* (MDR strain). They have reported that a concentration of 20 µg/mL AgNPs had a 67% inhibitory effect on biofilm against sensitive strains of *P. aeruginosa* and a 56% inhibitory effect against antibiotic-resistant strains. This evidence suggests that AgNPs can be used as an alternative treatment against the MDR bacterial strain of *P. aeruginosa* [162].

Awadelkareem et al. studied the potential of biologically synthesized NPs in inhibiting QS-regulated virulence and biofilm formation. They performed this study against *C. violaceum* and *P. aeruginosa*. The AgNPs inhibited violacein and acyl homoserine lactone at 85.18% at 1/2 × MIC and 78.76% at 1/2 × MIC, respectively, and simultaneously in the case of P. auroginosa, reduction in pyocyanin activity, total protease, Las A, and Las B was observed. In addition, exopolysaccharide production by *C. violaceum* and *P. aeruginosa* was significantly inhibited [163].

Syrians et al. reported that AgNPs synthesis was carried out by in situ green reduction method, where AgNPs were capped with semi-synthetic polysaccharide-based biopolymer (carboxymethyl tamarind polysaccharide). UV, DLS, FE-SEM, EDX, and HR-TEM characterize these CMT-capped AgNPs. NPs show a particle size ~20–40 nm, with long-term stability, observed by their unchanged SPR and zeta potential values. NPs showed a significant level of inhibition. The activity was checked against both Gram-positive (*B. subtilis*) and Gram-negative bacteria (*E. coli* and *Salmonella typhimurium*) [164].

The study by Zhang et al. proposed using AgNPs as an alternative treatment for eradicating resistant bacteria. However, the exact mechanism of AgNPs against biofilm has yet to be cleared. However, this study demonstrated the molecular level of investigation to identify the mechanism of AgNPs against the MDR resistant strain of *Pseudomonas aeruginosa* through an anti-biofilm assay by SEM and TET-labeled quantitative proteomics. The antibiofilm activity of AgNPs against the antibiotic-resistant *P. aeruginosa* biofilm was detected by examining the biofilm matrix’s disruption and cell death. In SEM analysis, they observed that when bacteria were exposed to the AgNPs, the structure of *P. aeruginosa* biofilm was destroyed, significantly reducing its biomass. The TMT-labeled quantitative proteomic analysis showed that AgNPs could beat the P. aeruginosa biofilm. It hampers the biofilm in different ways, affecting adhesion and motility, stimulating oxidative solid stress response, down-regulation of iron hemostasis, obstructing aerobic and anaerobic respiration, and involving the quorum-sensing system. This finding puts light on the mechanism of AgNPs against biofilms by providing a theoretical basis for its clinical application [165].

Singh et al. describe the AgNP production by putative *Cedecea *sp. The strain was isolated from the soil. The isolate has potential application information of spherical, crystalline, and stable AgNPs characterized by UV-visible spectroscopy, TEM, DLS, and FTIR. They tested the activity of AgNPs against four pathogenic bacterial strains, namely *S. epidermidis*, *S. aureus*, *E. coli*, and *P. aeruginosa*. They observed intense minimum inhibitory concentration (MIC) values of 12.5 and 6.25 µg/µL and minimum bactericidal concentration (MBC) values of 12.5 and 12.5 µg/mL against *E. coli* and *P. aeruginosa*, respectively. One of the distinct features of the AgNPs produced by the *Cedecea* sp. extract was their extreme stability for an extended period. They possess solid antibacterial effects, demonstrate against *E. coli* and *P. aeruginosa* biofilms, and can anticipate continuing for extended periods [166].

#### 2.2.3. Application of Silver Nanoparticles in Different Sectors

Recent developments in nanotechnology include the use of nanoparticles in developing new and effective medical diagnostics and treatments. Here, we explain the different ways in which NPs are commonly used, namely wound healing, bone healing, dental applications, and other medical importance. We also highlight their applications in different sectors, such as wound healing, bone healing, cancer diagnosis, and other medical applications, along with the medicinal applications used in various sectors, such as food industries, textile, and waste management.

##### Wound Healing

Wound healing includes different stages of coagulation, inflammation, cellular proliferation, and complex matric and tissue remodeling. It has different pathways and immune cells’ regenerative functional skills. It also significantly impacts patients’ mortality, morbidity, and economic implications [167,168]. In the current clinical practices, prevention of wound dehiscence and infection at the wound is a major threat. [169] Generally, chemical or physical injuries induce cutaneous wounds, which may sometimes disturb the skin structure and functional integrity at different stages. Sometimes it may also cause permanent disability or death, depending upon the intensity of the injuries. [170] If the damages on the skin are deep different opportunistic pathogens, start infecting the wound, leading to pus formation. This is a significant issue in current healthcare practices [170,171,172,173].

There are a few representative biocomposites, namely ActicoatTM and BactigrasTM smith and Nephew(British Columbia), AqucacelTM (Conva Tec, Reading, UK), polyMem SilverTM (Aspen, CO, USA), and Tegaderm TM [171]. They have modified ionic silver and received approvals from the U.S. (FDA) for application on the wound dressing. Not only that, but commercial products are also being developed that have promising results. To enhance their efficiency, they are conjugated with naturally derived biomaterials such as modified cotton fabrics [172,173], bacterial cellulose [17,174], chitosan [38,175], and sodium alginate [176,177].

##### Bone Healing Mechanism of AgNPs

Worldwide, millions of people face an issue with bone dislocation, ocular pathogenesis, degenerative and genetic conditions, cancer, and fractures [178]. During the osseous tissues’ replacement, the chances of infection and bacterial colonization on those sites raise concerns. In addition, related infections are associated with high morbidity [179]. Bone is active tissue that undergoes regeneration and restoring processes by an intrinsic bone modeling mechanism [180]. The method of bone grafting is usually related to the replacement or fixing of significant defects that may affect the osseous tissues, such as tumors or traumas [181]. While performing these transplants, there are chances of getting infections that usually lead to high inflammation and loss of bone [182].

It was found that AgNPs improved MC3T3-1 pre-osteoblast cells and subsequent bone-like tissue mineralization compared to other NPs [76]. The bone regenerative process or self-repairing capacity limits during bacterial infection cause bone defects. AgNPs provide a broad-spectrum antimicrobial intrinsic effect as compared to the usual antibiotics. To avoid the uncommon bacterial resistance to AgNPs, they conjugated with other polymeric materials to provide a synergistic effect. Hydroxyapatite (HA) is a calcium phosphate salt used in crystalline form to prepare human bone, dentin, and dental enamel [183]. This material is extensively explored to provide osseous-related restorative and regenerative strategies, which can be used either in grafting or metallic implants as a coating material [184].

Despite performing surficial modification in various metal implant surfaces, silver can be integrated with HA in the metallic or ionic form [185]. They present a suitable choice as a fraction or bioactive and antimicrobial bone implant. AgNPs embedded with HA-based coating have proven their high efficiency against Gram-positive [186,187] and Gram-negative bacteria [181,188,189]. Nanosilver is usually used as a doping material in the bone replacement procedure for synthetic and bio-inspired bone scaffolds; recently, many studies have been reported [116,190]. Different experimental techniques were used to incorporate the nanosilver accurately in HA. This includes laser-assisted deposition, electrochemical deposition, magnetron sputtering, and ion-beam-assisted deposition sol-gel techniques [190]. Various studies also were carried out to understand acrylic cement’s clinical potential and feasibility with AgNPs. The acrylic cement modification with AgNPs was explored broadly, but it lacks material characteristics and mechanical properties [191,192,193,194].

It was also reported that to enhance the bone fracture healing process, AgNPs could promote the osteogenesis and proliferation of mesenchymal stem cells (MSCs) [164,195].

##### Other Medicinal Use of AgNPs

The healthcare and medicine sector took advantage of novel technologies, including nanotechnology, along with different potential antimicrobial efficiency of NPs. They can also be used in cosmetics, topical products, and catheter modification. In addition, various studies reported their anti-cancerous properties with novel pharmacological applications. The study proposes using AgNPs in facial masks to improve their protective ability. Y. Li et al. [196] investigated the facial mask coated with AgNPs and titanium dioxide. They found a 100% reduction in *E. coli* and *S. aureus* CFU. On similar bases, another study was performed on commercially available masks dipped or treated with AgNPs’ solution at a concentration of 50 and 100 PPM. Results show that the masks incorporated with AgNPs inhibited the growth of *E. coli* and *S. aureus* [197]. The results are so impressive that such masks can be used in hospital areas prone to high microbial contamination [198]. A recent study has shown the efficiency of a disinfectant with AgNPs incorporated as an active ingredient to decontaminate masks [195]. The formulation showed high effectiveness against *E. coli*, *S. aureus*, and *K. pneumonia*.

Catheter-coated with AgNPs is broadly studied to prevent infection and improve sterility. The scientist found that catheters coated with AgNPs or incorporated with NPs can inhibit biofilm formation for at least 72 h against *E. coli*, *S. aureus*, and *P. aeruginosa* strains. They also studied the possible toxic effect of this NPs coated catheter for 10 days and showed no toxic effect or inflammation. During this period, they found that 84% of NPs remained coated and stable during the study. This makes them safe and supports using these technologies to improve the sterility and safety of catheters with NPs incorporation [199,200,201]. However, more research on biocompatibility doses and release rate of AgNPs from catheters is needed to explore to prevent adverse effects. ON-Q silver soaterTM, silverlineR, and AgTive are a few commercially available products with varieties of impregnated catheters that varies from country to country.

##### Cancer Diagnosis

Different studies support the anticancerous activity of AgNPs. It was proven that NPs could be used as nanocarriers for drug delivery or cancer treatment. The analysis performed by Amin Awaselkareem et al. shows the potential anticancer, antibacterial, anti-quorum, and antibiofilm properties of AgNPs synthesized from Eruca sativa leaf extract. They examined the anticancer potential of NPs by MTT, scratch, and invasion assay. They carried out studies on human lung cancers cell (A549), and the result indicates that NPs inhibited the migration of A549 [163]. In recent years, photodynamic therapeutic approaches have been discovered for different purposes, such as diagnoses, treatment, and to prevent cancer. Because with fluorescent properties nanomaterials have remarkable uses in the field of diagnosis, the research of fluorescent NPs mainly concentrates on semiconductor particles, which are nothing but quantum dots [165,202].

On the other side, silver can also be easily detectable in visible spectra and is anticipated to have low toxicity. It can also easily prepare and have a high-stability solution [89,203]. They synthesized the fluorescent molecule by the facile photochemical method, which gives these particles long-term stability for a few minutes. To reduce Ag+ from silver trifluoroacetate in the presence of amines, they have used the photogenerated ketyl radicals method [204]. At last, they concluded that Lumine sense arises from particles supported by small metal clusters. The AgNPs showed plasma bonds in 390 to 420 nm in past work. Due to the small size of the nanocluster, they found some bands at 450 nm. Small particles smaller than 2 nm are usually called nanoclusters to distinguish between small particles [205]. In the past few years, silver has received great attention due to its high fluorescent intensity, which is more remarkable than Au, along with a notable biological application [206]. It has numerous applications in bioimaging [207], chemical sensing [208,209], fluorescence labeling [181], and single-molecule microscopy [187].

##### Nanosilver Applications in Other than Medicine

Water treatment: Nanosilver can purify water by killing bacteria and other pathogens. This makes it a potential alternative to chlorine, which can produce harmful byproducts when it reacts with organic matter [210,211,212].Food packaging: Nanosilver can be used as packaging materials that inhibit the growth of bacteria and other microorganisms, which can help extend the shelf life of food products [213,214,215].Textiles: Nanosilver can create antimicrobial clothing and other textiles, preventing bacterial contamination and odors [181,216,217].Electronics: Nanosilver can create conductive materials for use in electronic devices, such as computers and smartphones. It can also be used as printed circuit boards and other electronic components [218].Energy production: Nanosilver can be used to produce solar cells and fuel cells, which are used to generate electricity [219,220,221,222].

It is important to note that the use of nanosilver in some of these applications is still in the research and development stage, and further study is needed to determine the safety and effectiveness of these products.

#### 2.2.4. Drawbacks

The biomedical application of nanomaterial can be infused by their potentially hazardous effects on organs system in the body, which has been gradually increasing [206,223,224]. Therefore, it is time to evaluate the dynamic of AgNPs in vivo. Via different modes of administration of AgNPs, they are distributed into other organelle systems in a body [206,223,224]. It includes inhalation, ingestion, skin contact, and intravenous or subcutaneous injection. It goes through many systems, namely the dermis, respiratory tract, urinary, nervous, digestive, spleen, immune, and reproductive systems. The studies revealed that the most affected organelles are the spleen, liver, kidney, and lungs. It is also observed that few depositions are on teeth and bones. Small NPs can induce potential cytotoxicity and easily cross biological barriers such as blood-brain barriers.

NP is not only transported through directly exposed tissues but also the blood to different vital organs. The non-specific transportation can cause dermal toxicity, ocular toxicity, respiratory toxicity, and neurotoxicity, which may limit the AgNPs’ activity. This characteristic plays an essential role in the cytotoxicity of NPs depending on the mode of administration and features of NPs. To understand cellular uptake, accumulation, degradation, chemical transformation, and removal of AgNPs, Wang et al. [225] performed TEM and integrated synchrotron radiation-beam transmission X-ray microscopy (SR-TXM) with 3D tomographic imaging. The cellular biochemical changes may include cytotoxicity due to the transformation of AgNPs, i.e., Ag0 transformation into Ag+, Ag-O-, and Ag-S species. However, there is inadequate information regarding cytotoxicity and long-term adverse health effects [71].

The CDC report shows that the U.S. produced 20 tons of AgNPs in 2010 [226,227,228]. While worldwide, it is estimated that 450–542 tons of nanomaterial production [227], recent studies show that in 2018 the total nanomaterial production globally was around 5.5 to 100.000 tons annually [229]. The recommended exposure limit (REL) allowed is each 10 micrograms per cubic meter (µg/m^3^) as an hours’ time weight average (TWA) concentration (total mass sample) of Ag. According to current occupational safety and health administration (OSHA), (PEL), (MSHA), (and NIOSH), to evolve AgNPs’ biocompatibility for medical applications, it is necessary to study their cytotoxicity systematically [227].

#### 2.2.5. New Approaches of Silver Conjugated with Peptides, Antibiotics, Bioactive Agents, and Dendrimers

Due to the newly emerged cogent problem of MDR bacteria, it is necessary to find a novel treatment in the presence of AgNPs that could improve the potency of the drug molecule without causing harm to the host system [119]. Therefore, additional research has been carried out with the help of different carrier molecules such as peptides, dendrimers, and other bioactive agents. Antimicrobial peptides (AMP) are used as an alternative to broad-spectrum antibiotics [230]. Due to their amphipathic nature, these peptides efficiently target microbial membranes. Peptides are naturally occurring, and their biocompatibility is higher than the other synthetic drug molecules. Although due to the rising vogue, the modification of these peptides is required for their therapeutic application. At the same time, the conjugation of peptides with nanoparticles is suggested to improve their potency [54,119,231,232,233].

Indrani has provided a new antibacterial system. This study suggested using abundantly available antimicrobial peptide conjugation with AgNPs. The cysteine residues are tagged at the terminal of peptides with AgNPs, not only to enhance the binding property but also to improve the antimicrobial potential against several pathogenic bacterial strains such as *K. pneumonia*. The bacteriostatic effect of cysteine-tagged nanoconjugates was obtained in the range of 5–15 µM compared to 50 µM for peptides without cysteine. They used 13C-isotope-labeled media to track the metabolic lifecycle of E. coli to give different perceptions of the system. Molecular dynamics analysis showed the pore formation in the membrane bilayer, which is mediated through a hydrophobic collapse mechanism suggesting that biocompatible nanomedicine is a potential alternative to conventional antibiotics [234].

In a new approach for the usage of AgNPs to avoid the development of resistance by pathogens, researchers found that it is time to get another line of treatment. In this scenario, the conjugation of AgNPs and dendron can be a promising strategy to combat antibiotic-resistant pathogens, thanks to dendrons’ antimicrobial activity and capacity to deliver antibacterial agents such as antimicrobial proteins (AMP), lysozyme, and bacteriophage endolysin [235]. Different antibiotics and AMPs bind to the dendrimers to improve the potency of the drug molecules against different MDR strains. PAMAM dendrimers were linked with penicillin via PEG linkers, which maintains its antimicrobial activity. The system was found to be very effective against *S. aureus.* The composition was created so that the penicillin would remain bound to the dendrimers and could effectively deactivate the close surrounding of bacteria [236]. Choi et al. tested dendrimers-based vancomycin conjugates, and these nanostructures bound were tested against vancomycin-resistant and non-resistant *S. aureus* with nano-molar affinity [237]. A recent study carried out by G. Jiang et al. showed that heterofunctionalization poly-(amido-amine) (PAMAM) dendrimers that were conjugated with vancomycin and further incorporated with AgNPs showed a significant reduction in bacterial colonies of*S.aureusstrains* [238].

## 3. Conclusions

In the review, we emphasized research performed on the role of AgNPs as a promising antimicrobial agent and, in brief, their mode of action. It also included different synthesis methods and their mechanism of action on MDR bacterial pathogens. If the size of particles is within 10–20 nm, the activity is due to the NP’s internalization and interactions with intracellular content. Nevertheless, when the size of NPs is more than 20 nm, the action is due to the silver ions. Studies proposed that the difference in antibacterial activity between Gram-positive and Gram-negative bacteria is associated with differences in the cell wall composition. It was observed that the AgNPs are less effective on Gram-positive bacteria due to a higher percentage of peptidoglycan content in the cell wall. Hence, it becomes challenging to deactivate the Gram-positive bacterial cells by AgNPs, while in the case of Gram-negative bacteria, they have a lipopolysaccharide layer that attracts the positive charge of AgNPs. Therefore, it is observed that AgNPs are highly efficient against Gram-negative bacteria compared with Gram-positive bacterial cells. The potent mechanism of NPs in the case of microorganisms depends on different physical factors, namely size, shape, and charge on molecules. AgNPs are conjugated with other biologically active components, peptides, and dendrimers to avoid the development of resistance, whereas antimicrobial compounds get conjugated with NPs. They are used as an alternative treatment for the existing antibiotics.

The last crucial point for researchers continuing the work under the same title needs to find the exact mechanisms of action of NPs, as they can alter the activity of protein synthesis and damage the DNA and generations of ROS. However, the exact mechanism behind all these activities has yet to be explored. In addition, it is necessary to find the resistance development pattern, which may help to improve the synthesis of NPs to avoid the hitch of bacterial resistance. Along with the mechanism of action, we observed that it is needed to standardize AgNPs’ synthesis and characterization. This can hinder the development of effective and consistent antimicrobial and antibiofilm agents based on AgNPs. Different studies have also observed that the AgNPs may not harm humans but have some adverse effects on aquatic environments (aquatic organisms and plants). Future research regarding the toxic effect of AgNPs is required. Although AgNPs have shown promising antimicrobial and antibiofilm agents, scaling up their production for practical use takes time and effort. In this review, we provided a concrete summary of past studies and a footway for new in-depth studies in areas such as the limited understanding of the mechanism of action, lack of standardization of synthesized NPs, potential toxicity, and challenges in scaling up production. Addressing the above-mentioned limitations could pave the way for more optimized development of AgNPs that can accelerate clinical studies.

## Figures and Tables

**Figure 1 microorganisms-11-00369-f001:**
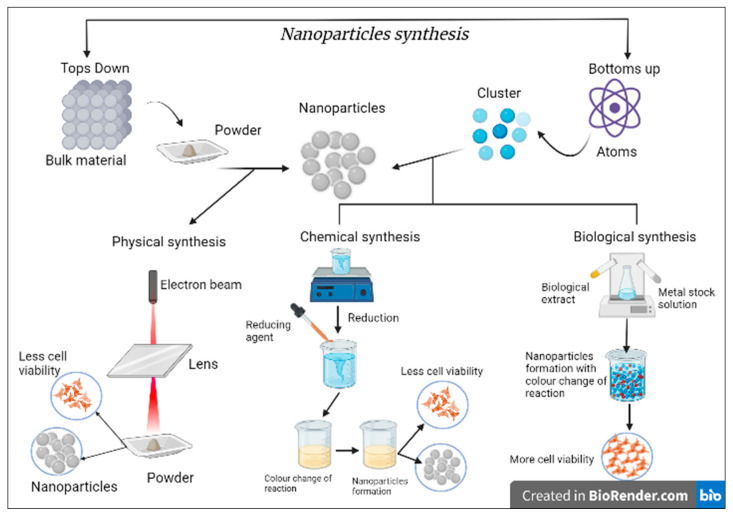
In brief, the AgNPs synthesis process by different synthesis approaches. The figure explains the top-down and bottom-up methods in detail. In the top-down method, it breaks down the bulk material into powder, and then this powder is used in the physical method of NP synthesis by laser ablation method. However, due to the toxicity, the cell viability is less. Likewise, in the bottom-up method, the atom form of the material is converted into a cluster. Then these clusters are used in synthesizing NPs by chemical or biological synthesis methods. In the biological method, cell viability is more than a chemical.

**Figure 2 microorganisms-11-00369-f002:**
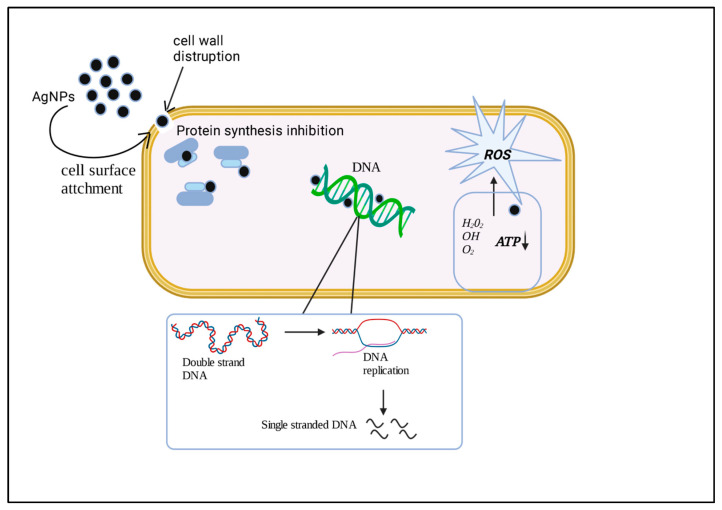
The figure explains the mechanical approach to NPs against the bacteria and the biofilm. The nanoparticles attach to the bacterial cell wall, penetrate the bacterial cell, and alter the different metabolic pathways. It affects DNA replication and protein synthesis; due to oxidative stress, ROS generation takes place. Furthermore, ultimately, it leads to cell death.

**Figure 3 microorganisms-11-00369-f003:**
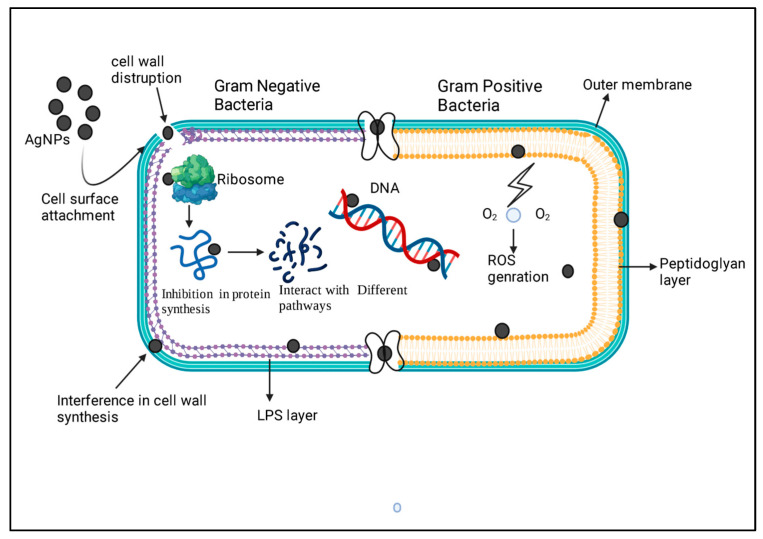
The comparative mechanistic approach of AgNPs on Gram-positive and Gram-negative bacteria surfaces has been explained. The left indicates the Gram-negative bacteria, while the right is a Gram-positive bacterial representation. The NPs will attach to the surface of the bacterial cell wall and depending on the cell wall composition, the NPs enter inside the bacterial cell. Furthermore, this leads to altering the other mechanistic approaches of the organism, likely protein synthesis, ROS pathway, and DNA replication.

**Figure 4 microorganisms-11-00369-f004:**
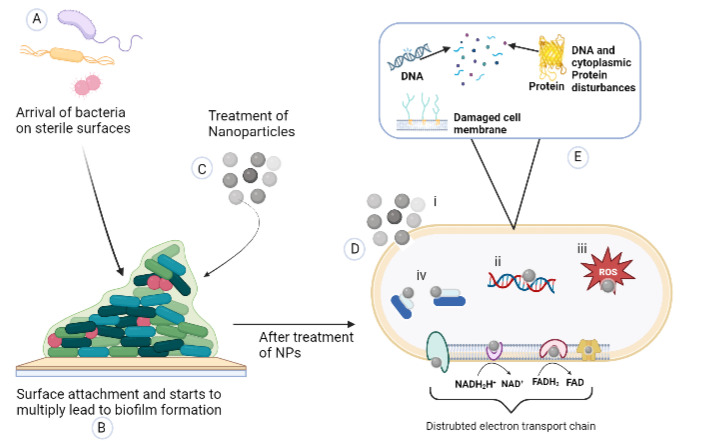
Diagrammatic representation of the activity of AgNPs on biofilm. (**A**) Bacterial cells approach biological or non-biological surfaces. (**B**) They adhere on the surface firmly and start multiplying, and the whole mass contains water, nuclei acid, polysaccharides, proteins, lipids, and ions (extracellular polymeric structures). (**C**) Treatment of NPs. (**D**) The NPs attack different vital functional sites of bacteria. (**E**) Shown inside of bacteria-like structure (**i**) Altered bacterial membrane shows electrostatic interaction. (**ii**) Damages the DNA and DNA replication with hydrophobic interaction. (**iii**) ROS generation by weak π-π interaction. (**iv**) Block the protein synthesis process through hydrogen bonding interaction. And causes disturbances in the electron transport chain by ionic interaction and van der Waal interaction.

**Table 1 microorganisms-11-00369-t001:** Antimicrobial activity of biologically synthesized NPs.

Sr. No	Biological Source (Plant Extract)	Tested Organism	Reference
1.	*Skimmia laureola*	*E. coli*, *K. pneumoniae*, *P. aeruginosa*, *P. vulgaris*, *S. aureus*	[35]
2.	*Erythrina abyssinica*	*E. coli and Salmonella*	[36]
3.	*Lysiloma acapulcensis*	*E. coli*, *S. aureus*,*P. aeruginosa*, *C. albicans*	[37]
4.	*Thymus vulgaris*, *Mentha piperita*, and *Zingiber officinale*	*Escherichia coli*, *Acinetobacter baumannii*, and *Staphylococcus aureus*	[38]
5.	*Endophytic bacterium Bacillus cereus*	*Escherichia coli*, *Pseudomonas aeruginosa*, *Staphylococcus aureus*, *Salmonella typhi* and *Klebsiella pneumoniae*	[39]
6.	*Origanum vulgare L*	*Shigella sonnei, Micrococcus luteus, Escherichia coli, Aspergillus flavus, Alternaria alternate, Paecilomyces variotii, Phialophora alba*	[40]
7.	*Rheum palmatum*	*Staphylococcus aureus* and *Pseudomonas aeruginosa*	[41]
8.	*Abelmoschus esculentus*	*Bacillus subtilis, Staphylococcus aureus*, *Staphylococcus epidermidis*, *Streptococcus pyogenes*, *Klebsiella pneumoniae*, *Escherichia coli, Pseudomonas aeruginosa*, *Proteus vulgaris*, *Salmonella typhimurium* and *Shigella sonnei*	[42]
9.	*Berberis vulgaris*, *Brassica nigra*, *Capsella bursa-pastoris*, *Lavandula angustifolia* and *Origanum vulgare*	*Staphylococcus aureus*, *Listeria monocytogenes*, *Escherichia coli*, *Salmonella enterica*, *Pseudomonas aeruginosa*	[43]
10.	*Impatiens balsamina* and *Lantana camara*	*Staphylococcus aureus* and *Escherichia coli*	[44]
11.	*Rowan Berries*	*P. aeruginosa* and *E. coli*	[45]
12.	*Carduus crispus*	*Escherichia* *coli*, *Micrococcus luteus*	[46]
13.	*Berberis Vulgaris*	*Escherichia coli* and *Staphylococcus aureus*	[47]
14.	*Syzygium aromaticum (clove)*	*Streptococcus mutans*, *Staphylococcus aureus* and *Enterococcus faecalis*	[48]
15.	*Rhodiola rosea*	*Escherichia coli UTI 89, and Pseudomonas aeruginosa PAO1*	[49]
16.	*Ligustrum vulgare berries*	*P. aeruginosa* and *E. coli*	[50]
17.	*Dried orange peel extract*	*Staphylococcus aureus (MRSA)*	[51]
**Sr. No**	**Biological Source** **(Microorganism)**	**Tested Organism**	**Reference**
1.	Arnicae anthodium	*Staphylococcus aureus*, *Escherichia coli*, *Pseudomonas aeruginosa,* and *yeast Candida* *albicans*	[52]
2.	Saccharomyces cerevisiae	*Staphylococcus aureus* and *Escherichia coli* ATCC 25922 test strains and *Staphylococcus aureus* 1536 and *Klebsiella pneumoniae* 520	[53]
3.	*Yeast strains HX-YS* and *LPP-12Y*	*P. aeruginosa*, *E. coli ATCC8099* and *S. aureus ATCC10231*	[54]
4.	Saccharomyces cerevisiae	*Staphylococcus aureus* and *Pseudomonas aeruginosa*	[55]
5.	*Rhodotorula* sp. *strain ATL72*	*Streptococcus *sp., *Bacillus *sp., *Staph *sp., *Shigella* sp., *Escherichia coli*, *Pseudomonas aeruginosa*, *Klebsiella* sp. and *Candida* sp.	[56]
6.	Penicillium oxalicum	*Staphylococcus aureus*, *S. dysenteriae*, and *Salmonella typhi*	[57]
7.	Penicillium diversum	*Escherichia coli*, *Salmonella typhi*, *Vibrio cholerae*, and *the clinical isolate Paratyphia*	[58]
8.	*Phenerochaete chrysosporium (MTCC-787)*	*P. aeruginosa*, *K. pneumoniae*, *S. aureus* and *S. epidermidis*	[59]

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
