# Peer review of "Silver Nanoparticles: Bactericidal and Mechanistic Approach against Drug Resistant Pathogens"

_microorganisms, 2023, doi:10.3390/microorganisms11020369_

Round 1

Reviewer 1 Report

THE REVIEW ARTICLE PRESENTS IN AN ADEQUATE MANNER THE STATE OF THE ART OF SILVER NANOPARTICLES, ITS BACTERICIDAL EFFECT AND THE MOST COMMONLY REPORTED AND ACCEPTED MECHANISMS OF ACTION SO FAR, AND THE EFFECT AGAINST BIOFILM FORMATION

INCLUDE A DISCUSSION REGARDING THE SHAPE OF SILVER NANOPARTICLES, ONLY THE SPHERICAL SHAPE IS DISCUSSED, WHICH IS THE MOST THERMODYNAMICALLY STABLE GEOMETRY, BUT THE SYNTHESIS OF OTHER GEOMETRIC FORMS OF NANOPARTICLES HAS ALSO BEEN REPORTED AND THEY HAVE DIFFERENT BACTERICIDAL EFFECTS

Author Response

General comment:

THE REVIEW ARTICLE PRESENTS IN AN ADEQUATE MANNER THE STATE OF THE ART OF SILVER NANOPARTICLES, ITS BACTERICIDAL EFFECT AND THE MOST COMMONLY REPORTED AND ACCEPTED MECHANISMS OF ACTION SO FAR, AND THE EFFECT AGAINST BIOFILM FORMATION

 Specific comments:

  1. INCLUDE A DISCUSSION REGARDING THE SHAPE OF SILVER NANOPARTICLES, ONLY THE SPHERICAL SHAPE IS DISCUSSED, WHICH IS THE MOST THERMODYNAMICALLY STABLE GEOMETRY, BUT THE SYNTHESIS OF OTHER GEOMETRIC FORMS OF NANOPARTICLES HAS ALSO BEEN REPORTED AND THEY HAVE DIFFERENT BACTERICIDAL EFFECTS.
  • Response:

As requested by the reviewer, we have provided the additional discussion. Kindly find the new information with detailed references on page 9, lines 247-267.

Reviewer 2 Report

Dear Authors, 

Though it is an interesting review article, which can be considered due to its importance in the field and problems associated to antibiotic resistance and biofilm infections in these times, but it needs lot of improvement before it should be considered. Please see my further comments below.

Firstly, it was very hard to read the manuscript, I started to fix some mistakes, but then I gave up as there is a problem nearly in each sentence; either the sentence is too short, the words used in the sentence are not the best choice or not used normally in that context, grammatically wrong or the author did not deliver the message (meaningless sentences), huge amount of typo errors, capital and Italics. Please revise the manuscript thoroughly for English language issues. Simple example is "gram". Gram is noun....G must be in Capital. Gram negative etc...

Abstract is not meaningful. The abstract should state briefly the purpose of the research, the principal objective in the review. What novelty authors brought from this review? and concluding remark. An abstract is often presented separately from the article, so it must be able to stand alone. In this manuscript, abstract looks more like an introduction and superficial general statements. It must be re-written as whole. 

Authors tried to bring the novelty in the manuscript by linking the problems of antibiotic resistance and biofilms and targeting through AgNPs. However, the background of the study is greatly lacking with a brief superficial passage on the development of AgNPs for biomedical applications. I did not find any novelty in the work at all. Initial paragraphs about antimicrobial activity of AgNPs, Antibiofilm activity of AgNPs, Biofilm etc all are well-known information. There is nothing new. Plethora of literature (10000s of papers on this subject) is available on web. Throughout the manuscript, authors have explained the details on regular aspect of antibiotic resistance, AgNPs synthesis procedure and possible mode of action. But the major aspect that is “Detailed Mechanism, success rate, real life applications in various sectors and fields" is completely missing from the review.

In my view, the authors have severely failed in providing a convincing context and research gaps present in the current literature or problems related to the subject and why their current study is valuable and/or deserving of the reader's time and attention. I highly suggest re-writing the background of the study in a more convincing manner with thorough explanation, apart from general info, which can also be removed. Focus on detail mechanistic studies. 

You have provided mechanistic figure for antimicrobial activity of AgNPs but not for antibiofilm. Design the figure for that too that how AgNPs are targeting biofilms and inhibiting biofilm. How AgNPs are targeting which proteins/pathways/EPS etc? Please design this figure which will give better impact to the study.

Authors have missed some important recent publications on AgNPs synthesized from natural sources and targeting antimicrobial, antibiofilm and anti-QS activities. Authors may consider taking some details and cite for better impact and add them in the tables.

https://www.mdpi.com/2079-6382/11/7/853

https://www.hindawi.com/journals/cjidmm/2022/9410024/

https://www.sciencedirect.com/science/article/pii/B978012821938600027X

There is no paragraph related to health implications of AgNPs, disadvantages etc. It must be added.

Prisma flow chart states n=200, which is very very low literature search number for survey methodology for a review article. This is not sufficient enough to present a novel review article.

Author Response

General comment:

Though it is an interesting review article, which can be considered due to its importance in the field and problems associated to antibiotic resistance and biofilm infections in these times, but it needs lot of improvement before it should be considered. Please see my further comments below.

Specific comments:

  1. Abstract is not meaningful. The abstract should state briefly the purpose of the research, the principal objective in the review. What novelty authors brought from this review? and concluding remark. An abstract is often presented separately from the article, so it must be able to stand alone. In this manuscript, abstract looks more like an introduction and superficial general statements. It must be re-written as whole.

  • Response:

As requested by the reviewer, the abstract has been completely rewritten to reflect the key points discussed in the paper. Kindly find the modified abstract on page 1, lines 16-32.

  1. Authors tried to bring the novelty in the manuscript by linking the problems of antibiotic resistance and biofilms and targeting through AgNPs. However, the background of the study is greatly lacking with a brief superficial passage on the development of AgNPs for biomedical applications. I did not find any novelty in the work at all. Initial paragraphs about antimicrobial activity of AgNPs, Antibiofilm activity of AgNPs, Biofilm etc all are wellknown information. There is nothing new. Plethora of literature (10000s of papers on this subject) is available on web. Throughout the manuscript, authors have explained the details on regular aspect of antibiotic resistance, AgNPs synthesis procedure and possible mode of action. But the major aspect that is “Detailed Mechanism, success rate, real life applications in various sectors and fields" is completely missing from the review. In my view, the authors have severely failed in providing a convincing context and research gaps present in the current literature or problems related to the subject and why their current study is valuable and/or deserving of the reader's time and attention. I highly suggest re-writing the background of the study in a more convincing manner with thorough explanation, apart from general info, which can also be removed. Focus on detail mechanistic studies.

  • Response:

Following the reviewer’s request, we modified the backgrounds section and another part of the study as suggested by the reviewers. We tried to address mechanistic approaches in detail throughout the manuscript. Please find the corrections on pages 1-2, from lines 35-71.

  1. You have provided mechanistic figure for antimicrobial activity of AgNPs but not for antibiofilm. Design the figure for that too that how AgNPs are targeting biofilms and inhibiting biofilm. How AgNPs are targeting which proteins/pathways/EPS etc? Please design this figure which will give better impact to the study.
  • Response:

As suggested by the reviewer, we have designed a new figure representing the antibiofilm mechanism of AgNPs. Kindly find the figure on page 16, lines 374-384.

  1. Authors have missed some important recent publications on AgNPs synthesized from natural sources and targeting antimicrobial, antibiofilm and anti-QS activities. Authors may consider taking some details and cite for better impact and add them in the tables. https://www.mdpi.com/2079-6382/11/7/853 https://www.hindawi.com/journals/cjidmm/2022/9410024/ https://www.sciencedirect.com/science/article/pii/B97801282193 8600027X

  • Response:

As kindly indicated by the reviewer, we have made respective changes, and we also cite the suggested articles. Kindly find the changes on page 6, table 1, pages 17 and 19, lines 407 to 413, and 536 to 541.

  1. There is no paragraph related to the health implications of AgNPs, disadvantages, etc. It must be added.

  • Response:

As suggested by the reviewer, we have added information regarding health implantation and its disadvantages. Kindly find the changes on pages 17 to 21, lines 444 - 605.

  1. Prisma flow chart states n=200, which is very very low literature search number for survey methodology for a review article. This is not sufficient enough to present a novel review article.

  • Response:

As indicated by the reviewer, we have added a new trend in the mechanism of AgNPs. Focusing on the main objective of the mechanism of action, we choose this article, and we have tried to describe it with the help of different figures and comparative studies. We also added their real-life applications along with mechanisms in the field of medicine. Kindly find the changes on pages 6-14 and 17 to 20, lines 162 to 288 and 447 to 578.

Reviewer 3 Report

The manuscript “Silver nanoparticles: applications in the field of bio-medicine” is suitable for publication in Microorganisms 

The authors get a review of biosynthesized silver nanoparticles. A good summary of the work on this topic with a focus on the mechanism of action of micro-organism particles with the aim of how to react to resistant bacteria.

 Specific comments: 

  1. Row 44. Word “Explain” should be “explain”.
  2. “These biobased reducing agents also reduce the toxicity of the NPs by forming the biological corona on their surface, which makes them an efficient candidate to be used in different medicinal applications.”- this is not true in every case. Please check and also give an opposite review with references. For example, one of the explanations about reduction mechanisms is in this article https://doi.org/10.1016/j.colsurfb.2017.09.031
  3. One paragraph is missing explaining how the reduction of silver occurs using extracts. What are the groups on the surface of polyphenols, flavonoids, etc., that participate in the process of silver reduction? 

 I believe that it should be published after the MINOR revision. 

Author Response

General comment:

The manuscript “Silver nanoparticles: applications in the field of bio-medicine” is suitable for publication in Microorganisms. The authors get a review of biosynthesized silver nanoparticles. A good summary of the work on this topic with a focus on the mechanism of action of micro-organism particles with the aim of how to react to resistant bacteria.

Specific comments:

  1. Row 44. Word “Explain” should be “explain”.

  • Response:

We have revised the manuscript intensively in accordance with the comments. Thus the suggested word is not anymore exist.

  1. These biobased reducing agents also reduce the toxicity of the NPs by forming the biological corona on their surface, which makes them an efficient candidate to be used in different medicinal applications.”- this is not true in every case. Please check and also give an opposite review with references. For example, one of the explanations about reduction mechanisms is in this.

  • Response:

As kindly indicated by the reviewer, we have changed the sentences along with the cross references. Kindly find the changes on page 4, lines 153-155.

  1. One paragraph needs to be added explaining how the reduction of silver occurs using extracts. What are the groups on the surface of polyphenols, flavonoids, etc., that participate in the process of silver reduction?

  • Response:

As suggested by the reviewer, we have added more points behind the mechanism of the biological synthesis of NPs, in the presence of different functional groups. Kindly find the changes on page 4, lines 120 -153. 

Reviewer 4 Report

The title of this review paper is “Silver nanoparticles: applications in the field of bio-medicine”.

The main content of the review provides a brief mechanism of action for MDR variants and the potential applications of AgNPs in MDR and biofilm-forming organisms. That is, a comprehensive review of the antibacterial properties of AgNP was conducted.

1. The title of the paper is too broad. It should be corrected according to the main content of the review.

2. The description of each figure is insufficient. In Figures 1, 2, 3 and 4, you must indicate exactly what you want to present.

3. The latest trends in the antibacterial mechanism of AgNPs should be added.

4. A synthetic mechanism for the biological synthesis of AgNPs should be suggested.

5. Limitations of antimicrobial and antibiofilm research on AgNPs should be presented in the conclusion section.

Author Response

General comment:

The main content of the review provides a brief mechanism of action for MDR variants and the potential applications of AgNPs in MDR and biofilm-forming organisms. That is, a comprehensive review of the antibacterial properties of AgNP was conducted.

Specific comments:

  1. The title of the paper is too broad, and it should be corrected according to the main content of the review.
  • Response:

We have changed the title of manuscript to “Silver nanoparticles: Bactericidal and mechanistic approach against drug resistant pathogens” in accordance with reviewer comment.

  1. The description of each figure is insufficient. In Figures 1, 2, 3 and 4, you must indicate exactly what you want to present.
  • Response:

Kindly indicate by a reviewer that we have put a more detailed description under all the figures. Kindly find the changes on pages 3-4, lines 81 to 88, for Figure 1. Figure 2. Kindly find the changes on page 7, lines 189 to 193. also, on page 6, from lines 173 to 187, we have explained the figure thoroughly.  In figure 3, page 8, lines 224 to 228, in-depth points can find on lines 213 to 240.

  1. The latest trends in the antibacterial mechanism of AgNPs should be added.

  • Response:

As indicated by the reviewer, we have already added the new trend of the mechanism of AgNPs, and we have tried to describe it with the help of different figures and comparative studies. We also added their real-life applications along with mechanisms in the field of medicine. Kindly find the changes on pages 6-14 and 17 to 20, lines 162 to 288, and 447 to 576.

  1. A synthetic mechanism for the biological synthesis of AgNPs should be suggested.

  • Response:

As kindly suggested by the reviewer, we added the mechanism behind the biological synthesis of AgNPs; kindly find the changes on page 4, lines 120 -153.

  1. Limitations of antimicrobial and antibiofilm research on AgNPs should be presented in the conclusion section.

  • Response:

As kindly suggested by the reviewer, we added the limitation of antimicrobial and antibiofilm research on AgNPs. Kindly find the changes on pages 21 and 23, lines 579 to 608, and 669 to 683.

Round 2

Reviewer 2 Report

Manuscript is significantly improved by the authors and now can be accepted in its current form. Through changes have been done. Well done!!. Some minor typo errors are there, please check thoroughly. Example gram positive, it should be Gram positive. G must be in capitals. Gram is noun, it's a name.

Author Response

Specific comments:

  1. Manuscript is significantly improved by the authors and now can be accepted in its current form. Through changes have been done. Well done!!. Some minor typo errors are there, please check thoroughly. Example gram positive, it should be Gram positive. G must be in capitals. Gram is noun, it's a name.
  • Response:

We would like to thank the reviewer for their kind suggestions. As indicated by the reviewer, we have corrected all the typo errors. Kindly find the changes in revised manuscript with the yellow highlights.
